# Evaluation of Illumina and Oxford Nanopore Sequencing for the Study of DNA Methylation in Alzheimer’s Disease and Frontotemporal Dementia

**DOI:** 10.3390/ijms26094198

**Published:** 2025-04-28

**Authors:** Lorenzo Pagano, Davide Lagrotteria, Alessandro Facconi, Claudia Saraceno, Antonio Longobardi, Sonia Bellini, Assunta Ingannato, Silvia Bagnoli, Tommaso Ducci, Alessandra Mingrino, Valentina Laganà, Ersilia Paparazzo, Barbara Borroni, Raffaele Maletta, Benedetta Nacmias, Alberto Montesanto, Roberta Ghidoni

**Affiliations:** 1Molecular Markers Laboratory, IRCCS Istituto Centro San Giovanni di Dio Fatebenefratelli, 25125 Brescia, Italy; lpagano@fatebenefratelli.eu (L.P.); afacconi@fatebenefratelli.eu (A.F.); csaraceno@fatebenefratelli.eu (C.S.); alongobardi@fatebenefratelli.eu (A.L.); sbellini@fatebenefratelli.eu (S.B.); bborroni@fatebenefratelli.eu (B.B.); 2Department of Biology, Ecology and Earth Sciences, University of Calabria, 87036 Rende, Italy; davide.lagrotteria@unical.it (D.L.); ersilia.paparazzo@unical.it (E.P.); alberto.montesanto@unical.it (A.M.); 3Department of Neuroscience, Psychology, Drug Research and Child Health, University of Florence, 50139 Florence, Italy; assunta.ingannato@unifi.it (A.I.); silvia.bagnoli@unifi.it (S.B.); benedetta.nacmias@unifi.it (B.N.); 4IRCCS Fondazione Don Carlo Gnocchi, 50143 Florence, Italy; 5Azienda Ospedaliero-Universitaria Careggi SOD Neurologia 1, 50100 Florence, Italy; tommaso.ducci1@edu.unifi.it (T.D.); alessandra.mingrino@unifi.it (A.M.); 6Regional Neurogenetic Centre (CRN), Department of Primary Care, ASP Catanzaro, 88046 Lamezia Terme, Italy; valelagana@gmail.com (V.L.); raffaelegiovanni.maletta@asp.cz.it (R.M.); 7Department of Clinical and Experimental Sciences, University of Brescia, 25123 Brescia, Italy

**Keywords:** epigenetics, methylation, sequencing, bisulfite, long-read, NGS, Illumina, Oxford Nanopore, Alzheimer’s disease, frontotemporal dementia

## Abstract

DNA methylation is a critical epigenetic mechanism involved in numerous physiological processes. Alterations in DNA methylation patterns are associated with various brain disorders, including dementias such as Alzheimer’s disease (AD) and frontotemporal dementia (FTD). Investigating these alterations is essential for understanding the pathogenesis and progression of these disorders. Among the various methods for detecting DNA methylation, DNA sequencing is one of the most widely employed. Specifically, two main sequencing approaches are commonly used for DNA methylation analysis: bisulfite sequencing and single-molecule long-read sequencing. In this review, we compared the performances of CpG methylation detection obtained using two popular sequencing platforms, Illumina for bisulfite sequencing and Oxford Nanopore (ON) for long-read sequencing. Our comparison considers several factors, including accuracy, efficiency, genomic regions, costs, wet-lab protocols, and bioinformatics pipelines. We provide insights into the strengths and limitations of both methods with a particular focus on their application in research on AD and FTD.

## 1. Introduction

The term epigenetics refers to the study of heritable changes in gene expression that do not involve alterations to the underlying DNA sequence. These modifications can influence how genes are turned on or off and are crucial for various biological processes, including development, differentiation, and diseases [1,2,3]. Epigenetic changes can be stable and heritable through mitosis/meiosis, without altering the DNA sequence itself [4]. Importantly, they can also be reversible, which has significant implications in treating diseases and developing novel therapeutic strategies [5]. Recent advances in epigenetics have highlighted its role in various diseases, including cancer, cardiovascular disorders, and neurodegenerative diseases [6,7,8,9,10]. Epigenetic mechanisms include DNA methylation, histone modification, and non-coding RNA interactions [6].

DNA methylation is a critical epigenetic process involving the addition of methyl groups to the C-5 position of the cytosine, forming 5-methylcytosine (5 mC). It has a role in many physiological processes such as the regulation of tissue-specific gene expression, genetic imprinting, X-inactivation, and transposons silencing to ensure genomic stability [11]. In mammals, DNA methylation mainly occurs in cytosines followed by a guanine nucleotide (CpG) and accounts for around 1% of the genome in healthy human tissues [4,12]. The majority of CpGs in the genome are hypermethylated while CpG-rich regions known as CpG islands (CGI), predominantly found at gene promoters, remain unmethylated [13]. CpG islands are short stretches of DNA where the frequency of CpG dinucleotides is significantly higher than in the rest of the genome. These regions are approximately 1 kb in length with a GC content greater than 50% and a ratio of CpG to GpC greater than 0.6 [14,15]. DNA methylation can modulate gene expression by inhibiting transcription when methyl groups are present in gene promoters. On the contrary, unmethylated CpG islands in promoters result in the transcription of a particular gene [16,17]. DNA methylation patterns are established and regulated by a family of enzymes called DNA methyltransferases (DNMTs). DNMT1 maintains the DNA methylation in somatic cells while DNMT3a and DNMT3b perform de novo methylation during development [18,19]. Active DNA demethylation is mediated by enzymes from the ten-eleven translocation (TET) family, which oxidize 5 mC to form 5-hydroxymethylcytosine (5 hmC). Further oxidation steps produce intermediate products, ultimately leading to the erasing of DNA methylation [20,21].

DNA methylation is highly tissue-specific and varies substantially between cell types. In the human brain, it is a key process for the proliferation and differentiation of neural stem cells, as well as in synaptic plasticity, learning, and memory. Alterations in DNA methylation patterns are associated with several brain disorders, including dementias such as Alzheimer’s disease (AD) and frontotemporal dementia (FTD) [22]. AD and FTD are two neurodegenerative dementias characterized by cognitive decline, leading to function disability [23]. They are considered multifactorial diseases, due to the complex interactions between genetic and environmental factors.

Genetic factors play an essential role in familial forms of both AD and FTD. In a very small proportion of cases, both diseases follow a Mendelian inheritance pattern with autosomal dominant transmission observed. High-penetrant mutations in three genes were identified to cause autosomal dominant AD (*APP*, *PSEN1*, and *PSEN2*), while causative, high-penetrant mutations in either one of the other three distinct genes (*MAPT*, *GRN*, and *C9orf72*) were identified as causative of monogenic FTD. Nevertheless, the majority of patients with AD and FTD appear to have a late-onset or sporadic form, with disease onset after the age of 65. In these cases, the interaction between genetic and environmental factors plays a crucial role in the multifactorial etiology of the disease [24,25,26]. Accordingly, neurotoxic agent exposures, alcohol consumption, smoking habits, or brain injuries can significantly raise the risk of AD and FTD by interfering with neurodegenerative genes through DNA methylation. Alterations in DNA methylation have been identified as key factors in the pathogenesis of these disorders [27,28,29,30,31,32,33,34,35,36].

Several epigenome-wide association studies (EWAS) conducted on AD and FTD patients revealed differentially methylated CpG sites (DMPs) and regions (DMRs), showing that specific DNA methylation alterations can influence the expression of genes involved in neurodegenerative processes [37,38,39,40]. DNA methylation changes have been observed in genes associated with the autosomal-dominant form of early-onset Alzheimer’s disease (EOAD), such as *APP* and *PSEN1*. Abnormal methylation of the *APP* gene can influence its expression and contribute to amyloid plaque formation, with increased methylation reducing gene expression [41,42]. In the case of the *PSEN1* gene, low methylation is associated with higher gene expression and disease progression [43]. Conversely, no studies have yet been found linking *PSEN2* methylation to AD. Altered methylation levels have also been found in genes associated with the autosomal dominant form of FTD, including *MAPT*, *GRN*, and *C9orf72*. Methylation of the *MAPT* gene is linked to an increased risk and severity of AD [44]; however, no significant differences in methylation levels have been found for FTD [39,45]. Specifically, in FTD, there is an increase in methylation of the *GRN* gene promoter, leading to a decrease in its expression [46]. Regarding the *C9orf72* gene, carriers of the repeat expansion exhibit significantly higher levels of methylation in the CpG island compared to non-carriers, with a clear association between expansion size, age of onset, and methylation status in FTD patients [47,48]. Different studies have identified various hypermethylated and hypomethylated loci in other critical genes associated with AD and FTD. There are strong evidences that ankyrin 1 (*ANK1*) gene is hypermethylated in AD cortex and involved in the neuropathology of the disease [49,50]. Brain DNA methylation differences have been found in genes classically associated with AD such as *SORL1*, *ABCA7*, *HLA-DRB5*, *SCL2A4*, and *BIN1* [51]. EWAS meta-analysis on FTD patients revealed differentially methylated loci in *OTUD4* and in *NFATC1*. Of these loci, *OTUD4* showed consistent upregulation of mRNA and protein expression in FTD [40].

Currently, various effective methods are employed for detecting DNA methylation, depending on the specific application, design of the study, or requirements. Array-based technologies represent a traditional approach that generates DNA methylation profiles for hundreds to thousands of samples at single-CpG resolution. Their main advantage is the economic efficiency, but this comes at the cost of a targeted design that interrogates only a small fraction of CpG sites across the genome.

For a more comprehensive understanding of epigenetic modifications that regulate gene expression and their impact on diseases, DNA sequencing-based approaches are essential [52]. When it comes to DNA methylation analysis, bisulfite sequencing technique is still considered the gold standard [53]. However, recent technological advancements have made the single molecule long-read sequencing technique an interesting alternative [54]. To date, a wide range of companies such as Illumina, Thermo Fisher Scientific, Roche, QIAGEN, PacBio, Oxford Nanopore, and BGI offer solutions for epigenetics research. Illumina confirms itself as the leader of the Next Generation Sequencing (NGS) market [55]. Considering the long-read sequencing (Third Generation Sequencing) segment, though, Oxford Nanopore accounts for the largest share [55].

No attempt has been made to compare these two technologies in the context of neurodegenerative dementias. This review aims to explore the advantages and limitations of Illumina and Oxford Nanopore sequencing techniques for DNA methylation analysis, and evaluate their application in research on neurodegenerative dementias such as AD and FTD.

## 2. Sequencing Techniques and Technologies

### 2.1. Illumina Bisulfite Sequencing

In bisulfite sequencing, the genomic DNA is treated with sodium bisulfite, resulting in a deamination of unmethylated cytosines to uracil and leaving methylated cytosines intact (Figure 1). The harsh treatment breaks the DNA into fragments; specific adapters are added to the ends of DNA fragments to assist in amplification and binding to the flow cell surface. PCR is then used to amplify the template, causing the conversion of the uracil to thymine. Subsequently, DNA is sequenced using short-read sequencing technologies like Illumina [56,57]. Illumina uses a sequencing-by-synthesis process, where DNA fragments bind to the flow cell and are locally amplified by PCR, creating clusters of identical molecules that enhance the reading signal. Reads are obtained by detecting fluorescent signals emitted by nucleotides added sequentially. Each sequencing cycle involves adding a nucleotide, detecting the fluorescent signal and removing the reversible terminator, allowing the next nucleotide to be incorporated [58]. Sequences are compared to a bisulfite-converted reference genome, and the percentage of methylation is estimated. As a result, 5 mC can be identified at single base-pair resolution using a qualitative, quantitative, and effective method [59,60]. Methylation studies using bisulfite sequencing can be employed through two methods: whole genome bisulfite sequencing (WGBS) and reduced representation bisulfite sequencing (RRBS). WGBS gives a full assessment of cytosine methylation covering most of the ~28 million CpGs in the human genome (≥2 CpG/100 bp); however, it remains the most expensive and resource-intensive technique [61,62,63]. As an alternative to WGBS, RRBS targets CpG-rich regions using a restriction enzyme that cuts at specific sites, allowing a reduction of costs and resources for the analysis. RRBS covers approximately 4 million human CpG sites [61,64,65].

### 2.2. Oxford Nanopore

Thanks to the advancement of long-read sequencing, it is now possible to detect methylation without inducing a chemical treatment like bisulfite conversion to the DNA (Figure 2). Long-read sequencing technologies can generate longer sequences (ranging from 10 kb to >1 Mb in length) compared to short-read approaches (150–300 bp), and they have the capability to directly sequence native DNA, allowing the identification of methylation from raw sequence data [66,67]. Oxford Nanopore (ON) uses an innovative method involving the passage of DNA molecules through a nanopore fixed in a synthetic membrane. Unlike NGS sequencing methods, ON does not require PCR amplification or DNA fragmentation, thereby preserving the native structure of long DNA strands. Specific adapters are attached to the DNA to facilitate its passage through the nanopores. The membrane of each nanopore is exposed to an electrical current, and as the DNA moves through the pore, variations in the electrical signal are detected. These signal variations are translated into nucleotide sequences using advanced computational algorithms. Nanopores are sensitive enough to distinguish electric shifts between methylated and unmethylated cytosines, enabling detailed mapping of methylation profiles on a large scale [66,68]. The difference can be determined by comparing the electric current pattern to an in silico reference or an unmodified control, or through pre-trained models, such as neural networks [69], machine learning [70], or Hidden Markov Models (HMM) [71]. However, a key challenge arises in detecting methylation at closely CpG sites, as it is often assumed that all CpGs within a 10-bp region share the same methylation status. The nanopore sequencer can sequence the whole genome (WGM) or select molecules in real-time, determining whether to continue sequencing or discard them based on predefined regions of interest. Therefore, it is possible to design target genomic regions such as CpG islands or CpG rich promoters, essentially achieving reduced representation methylation sequencing (RRMS), which is conceptually similar to the commonly used RRBS [72].

## 3. Comparison

### 3.1. Accuracy and Efficiency

In this context, accuracy refers to how precisely the sequencing output reflects the fidelity of both nucleotide identification and methylation detection. Efficiency, on the other hand, encompasses factors such as the alignment success rate and the number of CpG sites detected. A comparative genome-wide DNA methylation study across sequencing technology showed that the quality scores above Q20 and Q30 on Illumina NovaSeq 6000 are 92% and 83% for RRBS, 95% and 90% for WGBS [73]. However, bisulfite treatment is a highly aggressive method. This chemical process can induce a significant degradation of DNA, increasing the probability of base-calling error and leading to lower alignment rates and data loss. This may result in variance in methylation levels due to technical biases [74,75]. Several tools have been developed to enhance efficiency in bisulfite sequencing data analysis. A comprehensive benchmarking study compared 14 alignment algorithms, concluding that BSMAP achieved the highest efficiency for detecting CpG sites and methylation levels, with an alignment rate exceeding 90% and approximately 26.5 million CpG sites detected [76]. On the other hand, ON sequencing using R9.4.1 pore version and the Guppy version 6.1.2 in super accuracy mode for base and methylation calls demonstrated a raw read accuracy of almost Q20 (98.81%) and a median read identity mapped to the reference genome of 96%. Methylation calls in ON covered 28.83 million CpG sites, representing 98.8% of the total CpG sites [77,78]. Interestingly, a recent study comparing ON and Illumina bisulfite sequencing found a high correlation (r = 0.967) between the methylation levels detected by the two platforms. This suggests that despite differences in sequencing chemistry, both platforms provide reliable and qualitative insights into methylation patterns [79]. Nonetheless, it has been observed that ON is able to reach a much higher confidence in CpGs detection at a much lower depth than bisulfite whole-genome sequencing [78]. Additionally, bisulfite sequencing has a known limitation in distinguishing between 5 mC and 5-hmC modifications, leading to an overestimation of 5 mC levels [80]. In contrast, ON allows for direct, real-time, and simultaneous sequencing and detection of both 5 mC and 5 hmC modifications [68].

### 3.2. Genome Regions

Illumina WGBS allows for comprehensive genome-wide DNA methylation analysis at single-nucleotide resolution. This approach covers key genomic regions, including promoters, introns, exons, as well as 5′ and 3′ untranslated regions (UTRs). Alternatively, Illumina RRBS can be employed to enrich for CpG-dense regions, particularly CpG islands found in gene promoters. However, highly repetitive regions of the genome remain difficult to sequence and characterize using short-read sequencing technologies like Illumina, due to limited sequencing depth and poor mapping accuracy [81,82]. In the human genome, two predominant types of repeat elements are tandem repeats (TRs), which range from a single base to megabases, and interspersed repeats, primarily consisting of transposable elements (TEs) [82]. The advent of ON has enabled the resolution of many of these previously inaccessible “dark loci”, thanks to the longer read lengths it produces. As a result, new methylation patterns within repetitive DNA, including those associated with genes linked to neurodegenerative dementias, can now be identified.

### 3.3. Costs

The cost per sample for WGBS is relatively high, about 300 $ per sample depending on the required coverage and the option of paired- or single-end sequencing. This makes the technology more suitable for large-scale studies or projects requiring a comprehensive view of genomic methylation. RRBS is a more economical technique, with costs ranging from $100 to $300 per sample, significantly reducing expenses for studies with a large number of samples or for analyses focused on specific genomic regions [64].

The cost per sample with ON is variable and higher compared to Illumina, ranging from $500 to $1500 per sample, depending on the required coverage and the desired read length. Although it may seem expensive initially, the ability to monitor sequencing progress in real-time can reduce overall costs if sufficient coverage is quickly achieved.

### 3.4. Wet-Lab Protocols

Regardless of the sequencing methods employed, all wet-lab workflows begin with DNA extraction from the biological samples. WGBS requires at least 100 ng of high-quality DNA. To ensure reliable results, the input DNA should be intact and minimally fragmented, as bisulfite treatment can cause additional fragmentation. Before sequencing, fragments should have a length of 200–400 base pairs [83]. RRBS requires less DNA than WGBS, with a minimum amount of 50 ng. Although it focuses only on specific portions of the genome (CpG-rich regions), high DNA quality is still necessary [64,84].

ON requires a larger amount of DNA compared to Illumina, with 1 µg needed to obtain high-quality reads. It is crucial that the DNA is highly intact and not fragmented to ensure long and accurate reads, with DNA fragments exceeding 30 kb. Samples must follow specific purity requirements; contaminants such as salts or proteins can compromise sequencing quality and can lead to a reduction in the measured A260/280 and A260/230 ratios (required 260/280 ratio of 1.8 and 260/230 ratio of 2.0–2.2) [85]. Regarding the library preparation, both Illumina and ON protocols include steps for adapter ligation and barcoding. However, as previously mentioned, Illumina bisulfite sequencing has the disadvantage of requiring two additional steps which can introduce several errors and biases: the sodium bisulfite tratment and PCR amplification. Furthermore, the preservation of bisulfite-treated samples is crucial for maintaining DNA integrity. It is recommended to store converted DNA at −80 °C to prevent degradation, as inadequate temperatures can cause DNA instability due to the conversion of cytosines to uracils. Although converted and purified DNA can be stored at −20 °C for at least 9 months without significant quality loss, for longer periods it is preferable to store it at −80 °C [86]. The two workflows are summarized in Figure 3.

### 3.5. Bioinformatics Pipelines

The bioinformatics pipeline for bisulfite sequencing is almost identical for both WGBS and RRBS experiments [87,88]. Shafi et al. [89] provided a detailed overview of this multi-step process, with a particular focus on the many tools that can be employed at each step. The workflow can be summarized in four main phases: data preprocessing, alignment, methylation calling, and differential methylation analysis (DMA). Data preprocessing begins with a quality check of the short reads (in FASTA/FASTQ format), usually performed with the FastQC (http://www.bioinformatics.babraham.ac.uk/projects/fastqc/ (accessed on 24 April 2025)) program. Low-quality bases and adapters are then trimmed to improve alignment efficiency. There are many trimming tools that can be employed for this task, such as Cutadapt [90], Trimmomatic [91], and Trim Galore! (https://github.com/FelixKrueger/TrimGalore (accessed on 24 April 2025)). After preprocessing, reads are ready for alignment to the reference genome and methylation calling, which estimates methylation levels. These two steps can be performed with specific bisulfate sequencing aligners. Three-letter aligners—such as Bismark [92], BS Seeker [93], Bwa-Meth (https://github.com/brentp/bwa-meth (accessed on 24 April 2025)), and BRAT [94]—convert all Cs into Ts in the forward strand and all Gs into As in the reverse strand of the reference genome. Wildcard aligners (such as BSMAP [95], RRBSMAP [96], GSNAP [97], and BRAT-BW [98]) replace all the Cs in the reference genome with a wildcard letter Y instead, allowing it to match both Cs and Ts in the bisulfite sequencing reads. The methylation state of each cytosine in the reads is determined afterward. As with standard aligners, the output is a file in the SAM/BAM format. An optional post-alignment phase of quality control, methylation statistics, and visualization of methylated sites should be added for a more comprehensive analysis. Software tools like BiQ Analyzer [99] or aligners like BRAT can accomplish this. Finally, methylation levels between CpG sites are compared through DMA. A wide variety of hypothesis testing methods have been developed for this task. The classification proposed by Shafi et al. helps in bringing order to this vast array of approaches. Some of the existing methods use count values (the number of CpG sites or read counts) to test whether the difference in methylation between samples is significant or due to chance. Other methods use ratio values (such as methylation percentage) instead. Count-based approaches include those based on logistic regression (methylKit [100]), beta-binomial distribution (methylSig [101]), or Hidden Markov Models (ComMet [102]). Ratio-based approaches include methods based on smoothing (BSmooth [60]), entropy (QDMR [103]), binary segmentation (metilene [104]), or again, Hidden Markov Models (HMM-DM [105]). Some approaches, like COHCAP [106], integrate classical statistical tests such as Fisher’s exact test, t-test, and ANOVA. An accurate evaluation of the advantages and disadvantages of each method is strongly recommended before making a choice. In particular, it is important to weigh crucial parameters such as spatial correlation between CpG sites, error control, and confounding effects, including read depth variance and biological covariation. For example, logistic regression-based approaches typically consider read depth but often ignore biological variation among replicates and spatial correlation. Moreover, some methods may be specific to a particular sequencing protocol, i.e., WGBS or RRBS. Regardless of the chosen method, the output of DMA will always yield a list of differentially methylated sites and/or regions (i.e., clusters of adjacent CpG sites).

The bioinformatics workflow for the detection of 5 mC on ON starts with the translation of raw electrical signal data into nucleotide sequences using base-caller softwares. The number of base-calling programs has grown and their efficiency has increased in the last years. Some of the most used base-calling programs developed by ON are Guppy (https://nanoporetech.com/document/Guppy-protocol (accessed on 24 April 2025)) and Flappie (https://github.com/nanoporetech/flappie (accessed on 24 April 2025)). Guppy is a general purpose base-caller available only to ON customers, while Dorado and Flappie are open-source software [107]. After a QC check, nucleotide sequences are aligned to the human reference genome. Minimap2 is a widely used aligner thanks to its capability to work with long genomic reads of hundreds of megabases in length [108]. The subsequent step is the methylation calling. Over the years, several different methylation-calling tools have been developed. Liu et al. assessed the performance of different methylation-calling tools, concluding that Guppy and Nanopolish (https://github.com/jts/nanopolish (accessed on 24 April 2025)) are the best options for users with limited computational resources, while Megaladon (https://github.com/nanoporetech/megalodon (accessed on 24 April 2025)) showed better performances at the cost of high-demanding resources [109]. Base-calling, alignment to a reference genome, and detection of modified bases can also be performed using Dorado (https://github.com/nanoporetech/dorado (accessed on 24 April 2025)) as a single tool.

Regarding DMA, very few softwares are specifically designed for ON, although existing methods designed for bisulfite sequencing can be used. Recently, it has been demonstrated that pycoMeth, a toolbox for differential methylation for long-read sequencing data, performs comparably to or better than previous tools for the segmentation and further identification of differential methylated regions (DMRs) [110].

The great majority of available software tools for bioinformatics data processing are specifically designed for bisulfite sequencing. Some of these tools can also be employed for ON sequencing, but results should be carefully evaluated. ON-specific tools, despite being few, are often a better option. Regardless of the tools used, BS and ON sequencing pipelines are quite similar. However, a few differences can be highlighted.

ON requires an additional base-calling step to obtain FASTQ files, whereas Illumina outputs them directly. The adapter/quality trimming step is a key step in the BS pipeline, where well-documented and benchmarked tools are usually employed. In contrast, for ON there is no clear consensus on how to address this task. The effectiveness of the trimming step in reducing errors for ON sequencing is also in question [111]. Among the few available ON-specific trimming tools are Porechop (https://github.com/rrwick/Porechop/ (accessed on 24 April 2025)), Porechop_ABI [112], ProwlerTrimmer [113], Chopper (https://github.com/wdecoster/chopper (accessed on 24 April 2025)), and fastplong (https://github.com/OpenGene/fastplong (accessed on 24 April 2025)), the new fastp-based tool optimized for long-read data. Some aligners, such as Guppy and Dorado, can also perform adapter trimming. The alignment step also differs between BS and ON sequencing. BS aligners are designed to deal with the bisulfite treatment of the DNA, which involves some kind of base conversion as described previously. Conversely, ON aligners do not modify reads or references but are optimized to deal with the higher error rates caused by ON technologies and by the inherent difficulties of long-read alignments. A key challenge in methylation calling using ON sequencing lies in accurately detecting methylation at closely spaced CpG sites, due to signal interference resulting from the overlapping influence of adjacent bases during sequencing [109], a difficulty not encountered in BS. Nanopolish explicitly assumes that all CpGs within a 10-bp window share the same methylation state, which can result in inaccurate predictions when this assumption is violated [114]. Although the extent to which Guppy and Megalodon address this specific issue remains unclear, Liu et al. demonstrated that their performance, like Nanopolish, declines in the presence of non-concordant methylation states at neighboring CpG sites. While Dorado’s performance in this context has not been comprehensively assessed, its reliance on training datasets with fixed ratios of methylated and unmethylated bases suggests a potential susceptibility to misclassification [115]. In contrast, the deep learning-based method DeepBAM has been shown to achieve improved model performance by incorporating contextual information from extended k-mer windows, enabling more reliable detection of complex, heterogeneous methylation patterns [116].

## 4. Discussion

DNA methylation is a fundamental epigenetic mechanism involved in various physiological processes. Alterations in this mechanism can potentially lead to severe dysfunctions and disorders. Several brain disorders, including dementias such as AD and FTD are associated with modified methylation patterns. DNA sequencing is one of the most commonly used methods for detecting DNA methylation. More specifically, two sequencing approaches are currently dominant for DNA methylation detection: sequencing of bisulfite-converted DNA and single-molecule long-read sequencing [117].

To determine the most suitable approach for AD and FTD methylation studies, we compared Illumina sequencing, which involves DNA bisulfite conversion, with ON sequencing, which employs single-molecule long-read technology. We evaluated these two platforms across several aspects, including accuracy, efficiency, genomic regions covered, costs, wet-lab protocols and bioinformatics pipelines (Table 1). Illumina-based bisulfite sequencing, particularly WGBS, shows high accuracy, typically producing a significant percentage of bases above Q20 and Q30 and with single-nucleotide resolution across the genome. On the other hand, the quality of ON long-read sequencing has made substantial improvements in accuracy over the past decade, now reaching the Q20 accuracy for the detection of 5 mC. Although the accuracy of Illumina bisulfite sequencing remains slightly superior, Oxford Nanopore has the advantage of directly detecting methylation without the need of bisulfite treatment, thereby preserving the DNA original structure and avoiding the degradation or amplification biases that may occur with bisulfite conversion and PCR amplification [118]. Both technologies demonstrate comparable efficiency in terms of alignment rate and number of detected CpG sites. Additionally, it has been shown a high concordance in methylation levels between ON and Illumina (r > 0.95). In terms of initial DNA quality and quantity, ON sequencing imposes more stringent requirements than Illumina. Cost considerations are also essential: ON is generally more expensive, making Illumina a more feasible option when resources are limited. Consequently, cost per sample should be a primary factor in study planning and experimental design. Another often-overlooked aspect when selecting the best suitable platform is the bioinformatics processing. Due to the recent emergence and growing adoption of ON technology, there are relatively few tools specifically developed for its data. Furthermore, these tools have not undergone the extensive testing and benchmarking that tools designed for Illumina platforms have. One major challenge in methylation calling using ON tools is the reliable identification of methylation at CpG sites that are located close together. While it has been suggested that closer neighboring CpG sites are more likely to share the same methylation status, with co-methylation decreasing as the distance between CpG sites increases [119], a more recent study has been shown that a substantial number of adjacent CpG loci in human genome display discordant and cell-specific methylation patterns, and they are involved in regulation of the activity of enhancers [120]. Disregulation of enhancer methylation in neurons has been associated with AD pathology and cognitive symptoms. A large cluster of significantly hypomethylated enhancers has been identified within the *DSCAML1* gene that targets *BACE1*. Hypomethylation of these enhancers in AD is associated with an upregulation of *BACE1* transcripts and an increase in amyloid plaques, neurofibrillary tangles, and cognitive decline [121]. Since DNA methylation landscapes vary significantly between brain regions and across different cell types [122,123,124], it is not well known whether assuming uniform methylation across adjacent CpG sites may obscure important regulatory regions and lead to oversimplified interpretations of epigenetic mechanisms underlying neurodegenerative disease processes. An experimental comparison of ON and Illumina in detecting DNA methylation at closely spaced CpG sites, particularly within enhancers, may highlight important differences and consequently guide the selection of the most appropriate technology for studies on neurodegenerative diseases focused on specific regulatory elements. Despite these limitations, nanopore sequencing offers a unique advantage by enabling the simultaneous detection of both 5 mC and 5 hmC modifications, a capability not achievable with bisulfite-based methods. This approach allows for more accurate real-time estimates of both modifications, eliminating the need for separate experiments that may alter or complicate downstream analyses. The 5 hmC has been observed to be enriched in the human brain, particularly in neurons, and have a role in neurological disorders [125,126,127,128].

During brain development and the early stages of adulthood, the 5 hmC accumulates, indicating a crucial role in the growth and function of neurons [129,130]. Different brain areas, including the entorhinal cortex, cerebellum, and neocortex, showed reduced 5 hmC levels in late-stage AD patients [131,132], while the temporal gyrus presented elevated 5 hmC levels [132,133]. However, a number of disagreements are still present, especially for temporal gyrus and hippocampus [134,135]. Smith et al. [136] demonstrated that hypohydroxymethylation had previously confounded estimates of DNA hypermethylation at the ANK1 gene in AD patients. As 5 hmC is a precursor of active DNA demethylation, its reduced levels may reflect a loss of this mechanism in AD. Global levels of 5 hmC have also been found to be altered in FTD patients, compared to controls [137], but no studies have expanded our understanding of which brain regions, cell types, or specific genes exhibit these differences. The lack of studies and the discrepancies indicate the need to amplify researches to pinpoint the precise function of 5 hmC in AD and FTD. To date, no studies have considered the possibility of studying 5 hmC in AD and FTD using nanopore sequencing. Most studies that have identified methylation differences in genes and regions related to AD and FTD have primarily used immunohistochemical techniques and microarray-based methods, which only interrogate a small fractions of CpG sites. There is still a significant need to elucidate DNA methylation patterns and provide conclusive results in the genomic regions implicated in these pathologies. WGBS or RRBS on Illumina platforms can provide a comprehensive methylation signature in relevant regions previously associated with AD and FTD, and potentially identify novel regions. Nevertheless, short-read sequencing platforms like Illumina face challenges in accurately sequencing highly repetitive regions of the genome, which can be overcome by ON sequencing. A direct long-read sequencing can effectively span large tandem repeats, providing insights into their length, nucleotide content, and associated base modifications [138]. In a recent study, Ramirez et al. [139] investigated the role and the impact of DNA methylation on repetitive regions in patients with AD, using ON sequencing. They identified differentially methylated promoters within repetitive/dark regions and alterations in bulk 5 mC modifications of alpha satellites at a chromosomal scale. These regions are not detectable with Illumina-based methods, but they could reveal processes such as inflammation and genomic instability in neurodegeneration [140,141]. The GGGGCC repeat (G_4_C_2_)*_n_* expansion in the *C9orf72* gene is the most frequently observed genetic cause of FTD. It has been shown that *C9orf72* gene expression is reduced in multiple brain regions of patients with the repeat expansion [142,143,144].

Although various studies have employed different methods to evaluate *C9orf72* promoter methylation, few have explored the potential of using long-read sequencing to better characterize epigenetic modifications in this gene. To our knowledge, only one study—by Udine et al. [145]—investigated the methylation levels in *C9orf72* using long-read sequencing, primarily in patients with amyotrophic lateral sclerosis (ALS). Their findings indicate that higher methylation levels are associated with longer repeat expansions, and that methylation levels correlate with age at disease onset.

To date, no studies have specifically explored the potential of using nanopore sequencing to investigate DNA methylation within *C9orf72* or to identify novel methylation signatures in repetitive regions in FTD patients. In summary, our review indicates that both technologies exhibit comparable efficacy in detecting DNA methylation; however, their suitability depends on the specific objectives and constraints of each study. Thus, we recommend a careful evaluation of each technology’s respective advantages and limitations when selecting the most appropriate platform for a given study. ON offers distinct advantages in assessing methylation patterns in AD and FTD, as it can target additional genomic regions beyond those accessible through Illumina bisulfite sequencing, including repetitive sequences potentially implicated in these pathologies. Furthermore, ON allows for the differentiation of 5 hmC, offering additional insights into its role in neurodegenerative dementias. Conversely, Illumina bisulfite sequencing may be more suitable for studies where DNA quality or quantity is suboptimal or when financial constraints demand a cost-effective approach. Illumina may also be preferable when the regions of interest are already known to be effectively sequenced through bisulfite sequencing.

## 5. Conclusions

This review compares Illumina bisulfite sequencing and ON sequencing for DNA methylation profiling, emphasizing the strengths and limitations of each technology, particularly in the context of AD and FTD research. The choice between Illumina and ON should be carefully considered, accounting for study objectives, sample quality, bioinformatics analysis, and budgetary constraints, as platform selection can significantly impact research outcomes. To date, few studies have directly compared these two technologies in their application to DNA methylation analysis, and none have focused specifically on neurodegenerative diseases. This review aims to serve as a preliminary guide for researchers investigating DNA methylation in AD and FTD through sequencing. However, our comparison provides only a general overview on the topic, and more targeted benchmarking studies are needed. Future research will also be essential to validate and expand ON’s application for detecting 5 hmC and methylation in repetitive regions, potentially unlocking new insights into the methylation landscape of AD and FTD. Choosing the appropriate platform could be a game changer in the discovery of novel epigenetic mechanisms that modulate the pathogenesis and/or progression of these diseases. Aberrant DNA methylation patterns that precede overt neurodegeneration may facilitate the early diagnosis of AD and FTD, particularly in sporadic forms. As epigenetics serves as a bridge between genes and the environment, identifying such patterns could also help clarify how environmental factors contribute to neurodegeneration. Lastly, given that DNA methylation is a reversible epigenetic modification, the development of novel compounds capable of inducing site-specific methylation or demethylation could enable precise therapeutic modulation of gene expression, offering new opportunities for the treatment of neurodegenerative disorders.

## Figures and Tables

**Figure 1 ijms-26-04198-f001:**
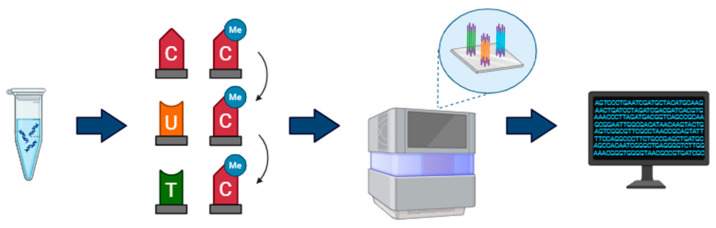
Illumina bisulfite sequencing (DNA; bisulfite treatment; sequencing; data analysis).

**Figure 2 ijms-26-04198-f002:**
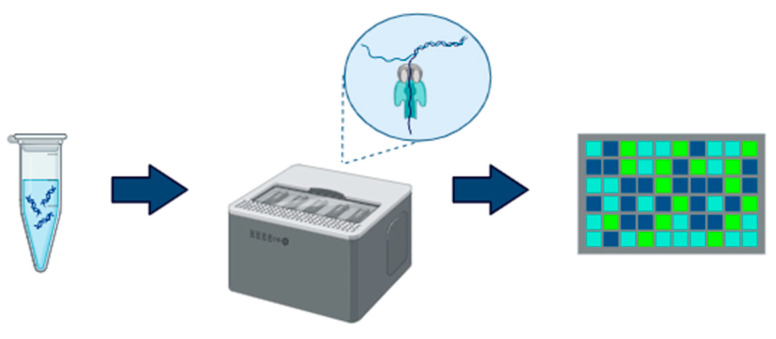
Oxford Nanopore sequencing (DNA; sequencing; data analysis).

**Figure 3 ijms-26-04198-f003:**
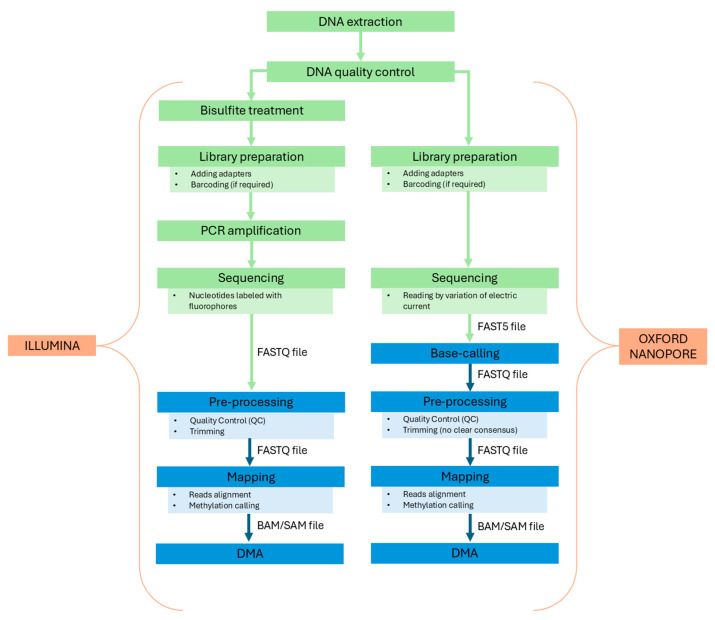
Workflow of Illumina bisulfite sequencing and Oxford Nanopore sequencing; green = wet lab; blue = bioinformatics pipelines.

**Table 1 ijms-26-04198-t001:** Differences between Illumina and Oxford Nanopore for accuracy, efficiency, genome regions, costs, wet-lab protocols, and bioinformatics pipelines.

Aspect	Illumina	Oxford Nanopore
Accuracy	Reaching Q20 and Q30; 5 hmC not detected	Almost Q20; real-time sequencing of 5 mC and 5 hmC
Efficiency	Alignment rate exceeding 90%; 26.5 million CpG sites	Alignment rate reaching 96%; 28.8 million CpG sites
Genome Regions	Repetitive regions are hard to sequence	Effective in resolving repetitive and dark genomic regions
Costs	WGBS~300$ per sample RRBS~200$ per sample	WGM ~ 1000$ per sample
Wet-lab protocols	need of bisulfite treatment and PCR; ~100 ng input; DNA fragments 200–400 bp	no need of bisulfite treatment and PCR; ~1 µg input; DNA fragments > 30 kb; 260/280 ratio of 1.8 and 260/230 ratio of 2.0–2.2
Bioinformatics pipelines	Specific tools with extensive testing and benchmarking	Few specific tools developed, especially for DMA

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
