# Peer review of "Evaluation of Illumina and Oxford Nanopore Sequencing for the Study of DNA Methylation in Alzheimer’s Disease and Frontotemporal Dementia"

_ijms, 2025, doi:10.3390/ijms26094198_

Round 1

Reviewer 1 Report

Comments and Suggestions for Authors

The manuscript titled “Evaluation of Illumina and Oxford Nanopore sequencing for the study of DNA Methylation in Alzheimer’s Disease and Frontotemporal Dementia” by Pagano et al offers a comprehensive and informative overview of current sequencing platforms applied in epigenetic research, with a particular focus on DNA methylation in Alzheimer’s Disease (AD) and Frontotemporal Dementia (FTD). The authors also provide a valuable primer on the epigenetic landscape of these neurodegenerative disorders, contributing to the contextual understanding of the technological applications discussed.

While the review is thorough and well-structured, further clarification of the manuscript’s novelty would strengthen its contribution. Specifically, the authors are encouraged to elaborate on how their comparative evaluation of Illumina bisulfite sequencing and Oxford Nanopore Technologies (ONT) long-read sequencing informs or advances studies in AD and FTD.

For instance, the inclusion of disease-specific considerations,such as brain tissue heterogeneity, methylation variability in distinct genomic regions, or platform-specific performance in complex tissues, would add practical depth. Summarizing these aspects in a comparative table, similar in format to Table 1 (page 11, lines 452–455), would enhance accessibility and impact.

Some further comments:

-Assumption of Uniform CpG Methylation: On page 5 (lines 187–189), the authors note: “However, a key challenge arises in detecting methylation at closely CpG sites, as it is often assumed that all CpGs within a 10-bp region share the same methylation status.”
This assumption warrants further discussion, especially in the context of brain tissue, where methylation profiles can exhibit substantial heterogeneity. The authors should address how this simplification may impact data interpretation in neurodegenerative disease research, and whether emerging computational or signal processing approaches may mitigate this limitation.

-In-text URL Formatting and Referencing: On lines 133–135 and in other instances throughout the text, URLs are embedded directly into the narrative. These should be replaced with properly formatted references in accordance with the journal’s citation style.

-Terminology: On page 3, line 146, the term “fragmentates” is used to describe the effects of bisulfite treatment. It is recommended that this be replaced with the more conventional term “fragments,” as the former is non-standard.

Reviewer 2 Report

Comments and Suggestions for Authors

This review provides a comparative overview of Illumina and Oxford Nanopore sequencing technologies for detecting DNA methylation, particularly in the context of Alzheimer’s disease (AD) and frontotemporal dementia (FTD). The topic is timely and of scientific significance. I would recommend acceptance after minor revisions, provided the following issues are addressed:

  1. The introduction would benefit from a brief overview of conventional techniques used for DNA methylation analysis, along with their advantages and limitations. This would provide a clearer background and context for the readers.
  2. Please improve the formatting of Figure 1 and Figure 2. Ensure that the font style and size are consistent across both figures. Additionally, elements such as test tubes and icons should be uniform in scale. The overall layout would be more effective if it were horizontally and vertically symmetrical.
  3. Out of the total 115 references, only 36 have been published since 2019, accounting for approximately one-third of the citations. To enhance the review's novelty and relevance, it is strongly recommended to incorporate more recent literature (from the past five years), particularly studies that apply or evaluate DNA methylation detection using Illumina or Nanopore platforms in neurodegenerative disorders.
  4. The Conclusion section lacks depth and critical reflection. As it stands, it provides only a superficial summary of the content without offering insightful interpretation or forward-looking perspectives. For example, to enhance the impact of this review, the authors are encouraged to expand the conclusion by discussing the implications of these technologies for future research or clinical applications.

Round 2

Reviewer 1 Report

Comments and Suggestions for Authors

The authors have sufficiently addressed the points raised.